

# A multifunctional GH39 glycoside hydrolase from the anaerobic gut fungus *Orpinomyces* sp. strain C1A

Jessica M. Morrison, Mostafa S. Elshahed and Noha Youssef

Department of Microbiology and Molecular Genetics, Oklahoma State University, Stillwater, OK, USA

## ABSTRACT

**Background.** The anaerobic gut fungi (phylum Neocallimastigomycota) represent a promising source of novel lignocellulolytic enzymes. Here, we report on the cloning, expression, and characterization of a glycoside hydrolase family 39 (GH39) enzyme (Bgxg1) that is highly transcribed by the anaerobic fungus *Orpinomyces* sp. strain C1A under different growth conditions. This represents the first study of a GH39-family enzyme from the anaerobic fungi.

**Methods.** Using enzyme activity assays, we performed a biochemical characterization of Bgxg1 on a variety of substrates over a wide range of pH and temperature values to identify the optimal enzyme conditions and the specificity of the enzyme. In addition, substrate competition studies and comparative modeling efforts were completed.

**Results.** Contrary to the narrow range of activities ($\beta$-xylosidase or $\alpha$-L-iduronidase) observed in previously characterized GH39 enzymes, Bgxg1 is unique in that it is multifunctional, exhibiting strong $\beta$-xylosidase, $\beta$-glucosidase, $\beta$-galactosidase activities ($11.5 \pm 1.2$, $73.4 \pm 7.15$, and $54.6 \pm 2.26$ U/mg, respectively) and a weak xylanase activity ($10.8 \pm 1.25$ U/mg), as compared to previously characterized enzymes. Further, Bgxg1 possesses extremely high affinity (as evident by the lowest $K_m$ values), compared to all previously characterized $\beta$-glucosidases, $\beta$-galactosidases, and xylanases. Physiological characterization revealed that Bgxg1 is active over a wide range of pH (3–8, optimum 6) and temperatures (25–60 °C, optimum 39 °C), and possesses excellent temperature and thermal stability. Substrate competition assays suggest that all observed activities occur at a single active site. Using comparative modeling and bioinformatics approaches, we putatively identified ten amino acid differences between Bgxg1 and previously biochemically characterized GH39 $\beta$-xylosidases that we speculate could impact active site architecture, size, charge, and/or polarity.

**Discussion.** Collectively, the unique capabilities and multi-functionality of Bgxg1 render it an excellent candidate for inclusion in enzyme cocktails mediating cellulose and hemicellulose saccharification from lignocellulosic biomass.

# INTRODUCTION

The production of biofuels from lignocellulosic biomass is a global priority, necessitated by the continuous depletion of recoverable fossil fuel reserves, the deleterious impact of fossil fuels on air quality, as well as their contribution to global climate change (*Hill et al., 2006*; *National Research Council, 2011*; *Ragauskas et al., 2006*). Lignocellulosic biomass represents

Corresponding author
Noha Youssef, noha@okstate.edu

a vastly underutilized and largely untapped source of energy, and its mass utilization for biofuel production is one of the goals enacted by the US Congress-implemented Renewable Fuel Standard (RFS), aiming to generate 16 billion gallons of biofuel from lignocellulosic sources by 2022 (*National Research Council, 2011*).

The most frequently used method of biofuel production from lignocellulosic biomass is the enzymatic conversion of cellulose and hemicellulose polymers into sugar monomers/oligomers that could subsequently be converted into biofuels using dedicated sugar metabolizers (*Elshahed, 2010*; *Hill et al., 2006*; *Kumar, Singh & Singh, 2008*). Historically, enzymatic cocktails designed for the breakdown of lignocellulosic biomass focused primarily on cellulose degradation, due to its relative structural simplicity and uniformity across all types of plant biomass. Nevertheless, the hemicellulose components in lignocellulosic biomass should not be ignored, as hemicellulose represents 20–35% of the composition of lignocellulosic biomass (*Liu, Saha & Slininger, 2008*). Unlike cellulose, plant hemicelluloses are structurally more complex, with multiple types of major hemicelluloses (arabinoxylans/glucuronoarabinoxylans, glucomannans/galactoglucomannans, mixed glucans, and xyloglucans) present in various plants (*Scheller & Ulvskov, 2010*). The most common type of hemicellulose are the arabinoxylans/glucuronoarabinoxylans that possess a structural backbone of $\beta$-1,4-linked xylose units (*Scheller & Ulvskov, 2010*). Xylan degradation requires the consorted action of the endo-acting-$\beta$-1,4-xylanases and the oligosaccharide depolymerizing $\beta$-xylosidases, among other enzymes (*Elshahed, 2010*; *Scheller & Ulvskov, 2010*).

The identification and characterization of novel enzymes and enzyme cocktails with superior lignocellulosic biomass saccharification properties (e.g., high substrate affinity and specific activity, activity retention at a wide range of pH and temperatures, and thermal and pH stability) signify essential thrusts in biofuel research. Members of the anaerobic gut fungi (phylum Neocallimastigomycota) represent a promising, and largely untapped, source of biomass-degrading enzymes (*Ljungdahl, 2008*; *Wang et al., 2013*). Members of the Neocallimastigomycota are found in the herbivorous gut, where they are responsible for the initial colonization and degradation of plant materials ingested by their hosts (*Ljungdahl, 2008*; *Wang et al., 2013*). While anaerobic gut fungi were initially discovered in sheep, they have since been found in the rumen and alimentary tracks of both ruminant and non-ruminant mammalian and reptilian herbivores (*Youssef et al., 2013*). The anaerobic gut fungi are excellent biomass degraders, capable of fast, efficient, and simultaneous degradation of the cellulolytic and hemicellulolytic fraction of various plants, including most common lignocellulosic biomass substrates (e.g., Corn Stover, Switchgrass, Sorghum, Energy Cane, and Alfalfa) (*Borneman, Akin & Ljungdahl, 1989*; *Harhangi et al., 2003*; *Liggenstoffer et al., 2014*; *Youssef et al., 2013*). Nevertheless, there have been extensive efforts dedicated to bioprospecting novel cellulases and hemicellulases from aerobic fungi (such as *Aspergillus* (*Kumar & Ramon, 1996*; *VanPeij et al., 1997*), *Trichoderma* (*Matsuo & Yasui, 1984*)), anaerobic prokaryotes (such as *Clostridium* (*Bronnenmeier & Staudenbauer, 1988*) and *Thermoanaerobacterium* (*Shao et al., 2011*)) and metagenomic sequence data (*Brennan et al., 2004*; *Hess et al., 2011*); in comparison to these numerous efforts, the identification, expression, and characterization of such enzymes from anaerobic fungi
has not been as well represented in the literature (*Borneman, Akin & Ljungdahl, 1989*; *Harhangi et al., 2003*).

We aim to explore the utility of the anaerobic gut fungus *Orpinomyces* sp. strain C1A (henceforth referred to as C1A) as a novel source of lignocellulolytic enzymes. C1A is an isolate from the feces of an angus steer on cellobiose-switchgrass media (*Youssef et al., 2013*). Our approach depends on implementing a transcriptomics-guided strategy to identify carbohydrate-active enzymes (CAZyme) transcripts that are highly expressed by C1A when grown on lignocellulosic biomass substrates as candidates for cloning, expression, and characterization. Here, we describe our efforts in cloning, expression, and characterization of one such enzyme: a GH39 transcript bioinformatically annotated as a β-xylosidase, designated Bgxg1, representing the first study of a GH39-family enzyme from anaerobic fungi. Our results document the high affinity, high specific activity, wide pH and temperature ranges, high thermal and pH stability of this enzyme, and novel multiple activities.

## MATERIALS AND METHODS

### Transcriptomics-guided selection of a GH39 enzyme for cloning and characterization

As a part of an extensive transcriptomic analysis of lignocellulosic biomass degradation by the anaerobic fungal isolate *Orpinomyces* sp. strain C1A (*Couger et al., 2015*), the most highly transcribed gene annotated as a β-xylosidase was selected for cloning and biochemical characterization. The selected m.21910 transcript (GenBank accession number KT997999) was annotated as member of the GH39 CAZyme family based on the presence of the conserved protein domain pfam01229 (Glyco_hydro_39) family. When strain C1A was grown on different substrates (glucose, Corn Stover, Energy Cane, Switchgrass, and Sorghum), m.21910 constituted 58–84% of the transcriptional activity (i.e., normalized FPKM values) of all GH39 transcripts ($n = 9$), and 5.7–18.2% of the transcriptional activities of all C1A genes putatively annotated as β-xylosidases (members of GH39 and GH43, $n = 41$) (*Couger et al., 2015*). The gene encoding for Bgxg1 protein was previously identified in the genome of strain C1A (GenBank contig accession number ASRE01002650.1, range: 2,346–3,460, see GCA_000412615.1 for whole genome). The ctg7180000059688.1 gene consists of 1,115 bp and no introns (refer to IMG gene ID 2518718918 for a visual representation of the gene, https://img.jgi.doe.gov/cgi-bin/m/main.cgi?section=TaxonDetail&page=taxonDetail&taxon_oid=2518645524). The protein product is predicted to be extracellular and non-cellulosomal, based on the presence of a signal peptide, and the absence of a CBM fungal dockerin domain, respectively.

### Bgxg1 sequence analysis and phylogeny

To determine the phylogenetic affiliation of Bgxg1 and the overall topology and global phylogeny of GH39 CAZymes, GH39 β-xylosidase sequences available in CAZY database (http://www.cazy.org/GH39_all.html) ($n = 1,145$ total GH39 sequences, retrieved October 28, 2015, edited to remove α-iduronidases and duplicates, resulting in $n = 200$ β-xylosidases), in addition to Bgxg1, were aligned using Clustal Omega (*Sievers et al., 2011*).

The generated alignment was used to construct a maximum likelihood tree in RAxML (*Stamatakis, 2014*), which was subsequently visualized and annotated using Mega6 (*Sievers et al., 2011*; *Tamura et al., 2013*).

## Synthesis, cloning, expression, and purification of Bgxg1 protein
### *bgxg1 gene synthesis and cloning*

A fraction (939 bp, positions 67–1,035) of m.21910 transcript was codon optimized for ideal expression in *E. coli* (see Fig. S1 for the alignment of the original gene and codon-optimized gene), and the *bgxg1* insert was synthesized by a commercial provider and inserted into a pET28a(+) plasmid (GenScript, Piscataway, NJ, USA). The plasmid, pET28a(+)-*bgxg1*, harbors kanamycin resistance (*kan*) and *NdeI* and *XhoI* restriction sites for selection and cloning. The pET28a(+)-*bgxg1* plasmid was first transformed into One-Shot Chemically Competent Top10 *E. coli* cells (Invitrogen, Carlsbad, CA, USA), and the transformants were grown overnight on LB-kanamycin agar (15 µg/mL) for selection. The purified plasmid was electroporated into a protease-deficient BL21(DE3)pLysS *E. coli* strain (Novagen, EMD Millipore, Darmstadt, Germany), possessing an additional chloramphenicol resistance (*cm*) marker, using a single pulse of 1.8 kV in 0.1 cm electrocuvettes. Transformants were grown on LB agar using both kanamycin (15 µg/mL) and chloramphenicol (34 µg/mL) for selection and screened for the presence of correctly sized inserts via colony PCR using T7 forward and reverse primers.

### *Bgxg1 expression and purification*

Ten milliliters of overnight cultures of BL21(DE3)pLysS *E. coli* cells transformed with pET28a(+)-*bgxg1* were used to inoculate 1 L LB broth, containing kanamycin (15 µg/mL) and chloramphenicol (34 µg/mL). The culture was incubated at 37 °C with shaking at 200 rpm until an $OD_{600} = 0.6$ was reached. Isopropyl-$\beta$-D-thiogalactopyranoside (IPTG, 1 mM final concentration) was then added to induce protein production, and the culture was gently shaken at room temperature overnight. Cells were then pelleted by centrifugation (6,000× g, 10 min, 4 °C) and the pellets were collected and stored at −20 °C.

Preliminary small-scale experiments indicated that the protein is expressed in the inclusion body fraction (Fig. S2). Inclusion body extraction was initiated by incubating the cultures in B-Per Cell Lysis Reagent (Thermo Scientific, Grand Island, NY, USA) (10 ml per 500 ml of culture) for 15 min at room temperature with gentle shaking to lyse the cells. The homogenate was centrifuged (10,000× g, 30 min, 4 °C) and the inclusion body extraction procedure (*Grassick et al., 2004*) was conducted on the cell pellet as follows: the pellet was resuspended in a urea-based inclusion body extraction buffer (20% glycerol, 8 M urea, 50 mM sodium monobasic phosphate, 500 mM sodium chloride, pH 8.0) for 30 min at room temperature with gentle shaking. The homogenate was centrifuged (10,000× g, 30 min, 4 °C) and the resultant supernatant containing target inclusion body proteins was subsequently utilized for refolding and purification procedures.

Recombinant protein refolding was achieved using slow dialysis as previously described (*Grassick et al., 2004*). In brief, inclusion body extract was incubated with EDTA (1 mM final concentration) and $\beta$-mercaptoethanol (100 mM final concentration) for 2 h at room temperature with gentle shaking, transferred to dialysis tubing (NMWL: 12,000–14,000 Da),

and placed for 3 h into inclusion body exchange buffer (20% glycerol, 8 M urea, 50 mM sodium monobasic phosphate, 500 mM sodium chloride, 1 mM EDTA, pH 8.0) for removal of the $\beta$-mercaptoethanol. The buffer was refreshed and dialyzed for an additional 3 h. The dialysis tubing was then placed into a low-urea refolding buffer (2 M urea, 50 mM sodium monobasic phosphate, 500 mM sodium chloride, 1 mM EDTA, 3 mM reduced glutathione, 0.9 mM oxidized glutathione, pH 8.0) and dialyzed overnight, followed by a no-urea refolding buffer (50 mM sodium monobasic phosphate, 500 mM sodium chloride, 1 mM EDTA, 3 mM reduced glutathione, 0.9 mM oxidized glutathione, pH 8.0) for 36 h.

Following dialysis, the contents of the tubing were centrifuged to remove insoluble, precipitated proteins (15,000× g, 15 min, 4 °C). The supernatant, containing refolded soluble protein, was then exposed to a nickel-nitriloacetic acid (Ni-NTA, 1:1 ratio) slurry (UBPBio, Aurora, CO, USA), packed in a glass frit column (25 × 200 mm, 98 mL volume Kimble-Chase Kontes Flex Column, Vineland, NJ, USA), and allowed to incubate at 4 °C for 1 h on an orbital shaker. Protein purification followed as detailed previously (*Morrison, Wright & John, 2012*). Samples were concentrated using Amicon Ultra-15 Centrifugal Filter Units (NMWL 30 kDa; Millipore) and protein concentration was determined using a Qubit Fluorimeter (Thermo Scientific) in reference to standard protein concentrations. Protein refolding was checked as activity against PNPX, as described below. An SDS-PAGE gel was run to check protein size and purity, as previously described (*Laemmli, 1970*; *Morrison, Wright & John, 2012*).

## Biochemical characterization of Bgxg1 (enzyme activity assays)
### pH and temperature optima and stability
The pH range and subsequent pH optimum for Bgxg1 was determined by assaying its $\beta$-xylosidase activity (described below) at pH 3, 4, 5, 6, 7, 8, 9, and 10, using the following buffer systems: sodium acetate buffer (pH 3.0–6.0), sodium phosphate buffer (pH 7.0–8.0), and glycine buffer (pH 9.0–10). Similarly, the temperature range and subsequent thermal optimum for Bgxg1 was determined by assaying its $\beta$-xylosidase activity at 25, 30, 39, 50, and 60 °C. In a second and separate study, the stability of Bgxg1 after exposure to pH extremes was determined by assaying its $\beta$-xylosidase activity following a one-hour incubation at pH 3, 4, 5, 6, 7, 8, 9, 10, 11, 12, and 13 at 4 °C. The following pH buffering systems were used for pH adjustment: sodium acetate buffer (pH 3.0–6.0), sodium phosphate buffer (pH 7.0–8.0), glycine buffer (pH 9.0–10), sodium bicarbonate (pH 11.0), and KCl–NaOH (pH 12–13). Similarly, in a separate study, the thermal stability of Bgxg1 was determined by assaying its $\beta$-xylosidase activity following a one-hour incubation at 4, 25, 30, 37, 39, 50, 60, and 70 °C. In all cases, 2.2 μg of pure Bgxg1 was used, since this concentration was determined to be optimal in initial testing. Following the one-hour long exposure at the above-described extremes, enzymatic activity was tested at 39 °C and pH 6.0, the optimal conditions as determined for this enzyme. All experiments were completed in triplicate, and relative specific activities in relation to the best performing condition (100% activity) were reported.

### Enzyme activity assays

All enzyme assays with Bgxg1 were conducted in pH 6.0 buffer and at 39 °C, as these conditions were determined to be optimal for Bgxg1. All reagents were purchased from Sigma Aldrich (St. Louis, MO, USA) unless noted otherwise.

Endoglucanase, exoglucanase, xylanase, and mannanase activities were determined using a DNS (3,5-dinitrosalicyclic acid)-based assay (*Breuil & Saddler, 1985*), with carboxymethyl cellulose sodium salt (CMC, 1.25% w/v), avicel microcrystalline cellulose (1.25% w/v), beechwood xylan (1.25% w/v), and locust bean gum (0.5% w/v) as substrates, respectively. Glucose, xylose, or mannose were utilized for the generation of a standard curve, dependent on the substrate being tested.

Cellobiohydrolase, $\beta$-xylosidase, arabinosidase, mannosidase, $\beta$-glucosidase, $\beta$-galactosidase, and acetyl xylan esterase activities were determined using (10 mM) of the *p*-nitrophenol-based (PNP) substrates: *p*-nitrophenyl-$\beta$-D-cellobioside (PNPC), *p*-nitrophenyl-$\beta$-D-xylopyranoside (PNPX), *p*-nitrophenyl-$\beta$-D-arabinofuranoside (PNPA) *p*-nitrophenyl-$\beta$-D-mannoside (PNPM), *p*-nitrophenyl-$\beta$-D-glucopyranoside (PNPG), *p*-nitrophenyl-$\beta$-D-galactopyranoside (PNPGal), and *p*-nitrophenyl-acetate (PNPAc), respectively (*Dashtban et al., 2010*; *Kubicek, 1982*; *Zhang, Hong & Ye, 2009*). Assays were conducted in sodium acetate buffer with sodium carbonate (1 M) as a stop reagent. The release of PNP was measured colorimetrically at 420 nm, following the addition of the stop solution. $\alpha$-glucuronidase activity was assayed using the Megazyme $\alpha$-glucuronidase assay kit (Wicklow, Ireland).

All experiments were conducted in triplicate. One unit of enzymatic activity ($U$) was defined as one µmol of products (reducing sugar equivalents in DNS assays, PNP released in PNP substrate-based assays, and aldouronic acid in $\alpha$-glucuronidase assay) released from the substrate per minute. Specific activity was calculated by determining the units released per mg of enzyme.

### Enzyme kinetics

Standard procedures were used to determine the $K_m$, $V_{max}$, and specific activity of Bgxg1 on all substrates described above (*Lineweaver & Burk, 1934*). $K_m$ and $V_{max}$ values were obtained using double-reciprocal *Lineweaver & Burk (1934)* plots, which were used to extrapolate from experimentally-derived values using a constant protein concentration (2.2 µg) and variable PNP-based substrate concentration (0.1–100 mM). Given the extinction coefficient of p-nitrophenol (PNP) is 17/mM/cm at 400 nm (*Bessey & Love, 1952*), for a 1 cm path length cuvette and absorbance minimum of 0.010, reliable $K_m$ detection limits in such PNP-based spectrophotometric assays is $\approx$500 nM. Therefore, $K_m$ values <500 nM are referred to as BDL (below detection limit).

### Substrate competition assays

Competitive inhibition experiments were conducted to determine whether the observed multiple oligosaccharide hydrolase activities are catalyzed via a single or multiple active sites. In such experiments, the effect of cellobiose (as a competitive inhibitor) on the $\beta$-xylosidase activity of Bgxg1 was measured by conducting the $\beta$-xylosidase assay, using 10 mM of PNPX as the substrate, in the presence of different concentrations of cellobiose

(0, 10, and 20 mM) and evaluating the impact of cellobiose presence on the release of PNP. Conversely, the effect of xylobiose (as a competitive inhibitor) on the $\beta$-glucosidase activity of Bgxg1 was measured by conducting the $\beta$-glucosidase assay (using 10 mM of PNPG as the substrate) in the presence of different concentrations of xylobiose (0, 10, and 20 mM), and evaluating the impact of xylobiose presence on the release of PNP. In both experiments, the effect of inhibitor concentration on $K_m$ and $V_{max}$ was evaluated using *Lineweaver & Burk (1934)* plots. All experiments were conducted in triplicate.

Substrate preferences of Bgxg1 were determined by conducting a substrate competition assay, where Bgxg1 (2.2 µg of pure enzyme preparation) was challenged by a mixture of xylobiose (10 mM) and cellobiose (10 mM). The kinetics of xylose and glucose release were compared to the results obtained in control experiments where only one substrate (xylobiose or cellobiose) was utilized. Samples were taken at 0, 1, 5, 10, 15, 30, and 60 min for the determination of the glucose and xylose concentrations. Glucose was assayed using PGO Enzyme Preparation Capsules (Sigma-Aldrich, St. Louis, MO, USA) and xylose was assayed using Megazyme Xylose Kit (Wicklow, Ireland). All experiments were conducted in triplicate.

### Bgxg1 modeling

Homology modeling by Iterative Threading ASSEmbly Refinement (I-TASSER) (*Roy, Kucukural & Zhang, 2010*; *Yang et al., 2015*; *Zhang, 2008*), was conducted to generate a three-dimensional model of Bgxg1 using *Thermoanaerobacterium saccharolyticum* $\beta$-xylosidase (PBD entry 1UHV) as a template. PyMOL was used to align the Bgxg1 structural prediction to that of *Thermoanaerobacterium saccharolyticum* (PBD entry 1UHV) to examine and speculate the impact of variations in amino acids residue on the enzyme's active site topology and putative substrate binding capacities (*PyMol, 2014*).

## RESULTS

### Bgxg1 phylogenetic affiliation

Phylogenetic analysis grouped all GH39 sequences into 4 phylogenetically-resolved and bootstrap-supported clades (Classes I–IV in Fig. 1). *Orpinomyces* sp. strain C1A Bgxg1 protein belonged to Class III, forming a well-supported cluster with GH39 proteins from the anaerobic fungus *Piromyces* sp. strain E2, as well as GH39 proteins from the bacterial genera *Clostridium* and *Teredinibacter* (70–74% sequence identities) (Fig. 1). To our knowledge, none of the GH39 proteins within this specific cluster, or in the entire Class III GH39, has been biochemically characterized.

### Physiological characterization

SDS-PAGE results show that the Bgxg1 protein is consistent with the predicted size of 42.7 kDa (protein predicted molecular weight is 39.6 KDa + 0.996 kDa linker + 2.101 kDa double histidine tag) (Fig. S3).

The thermal and pH ranges and optima were determined by conducting assays at a range of temperatures and pH's, as described above. Bgxg1 exhibited activity in a wide range of pH (3–8) and temperatures (25–60 °C), with optimal activity at pH 6 and 39 °C

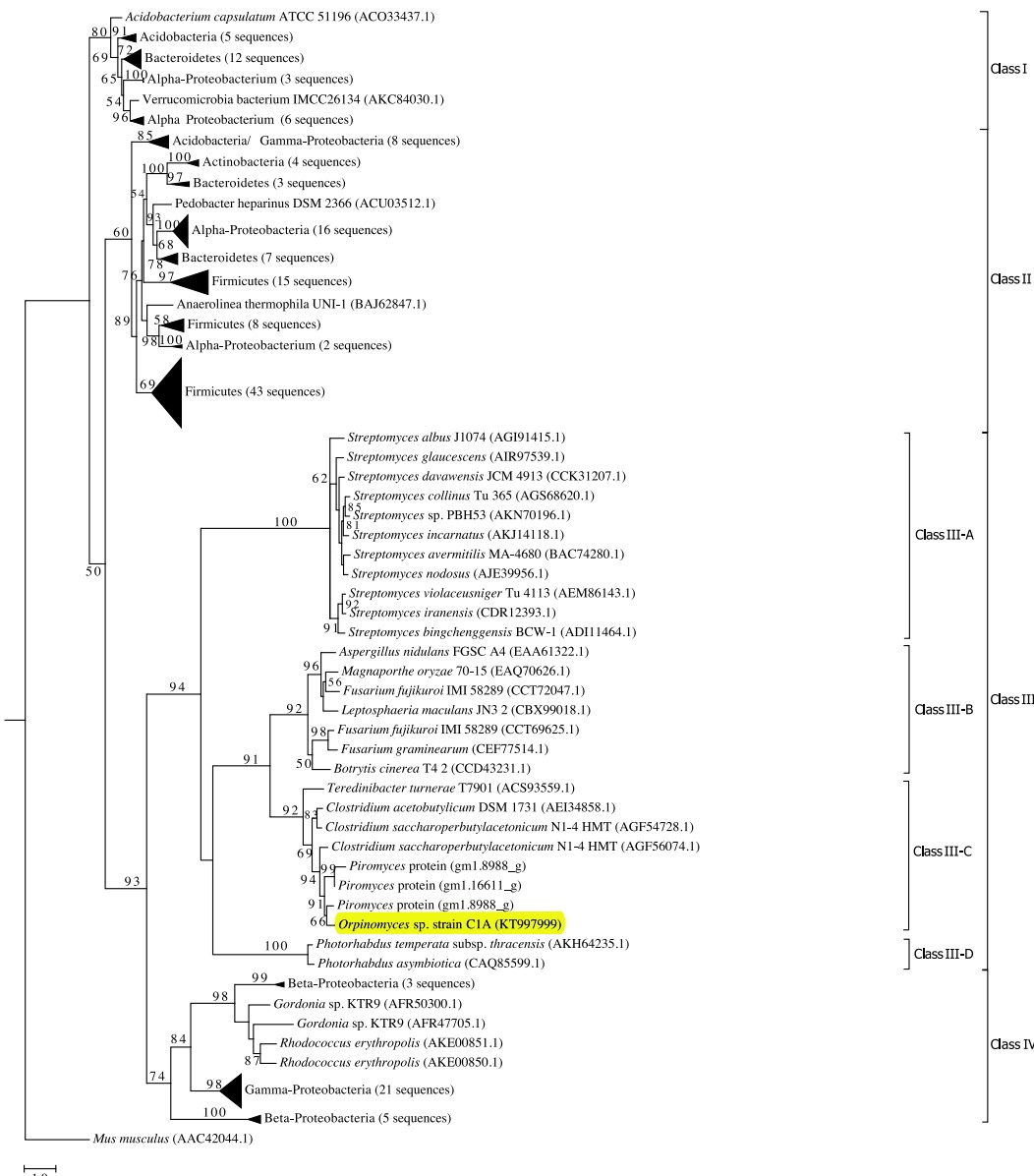

**Figure 1  Phylogenetic analysis of GH39 β-xylosidases, including Bgxg1.** Sequences annotated as GH39 β-xylosidases (*n* = 200 sequences, October 28, 2015) were retrieved from CAZyme databases (*Lombard et al., 2014*). Genbank accession numbers are shown for reference proteins (due to the unavailability of *Piromyces* proteins in Genbank, those proteins are shown as JGI accession numbers). The Maximum Likelihood tree was generated in RAxML (*Stamatakis, 2014*) using a BLOSUM62 substitution matrix and a GAMMA model of rate heterogeneity. The model estimated an alpha parameter of 2.069. Bootstraps values (100 replicates) are shown for nodes with >50 bootstrap support. The sequences were empirically classified into four classes (Classes I–IV), and Class III, to which Bgxg1 is affiliated, is further classified into four distinct lineages (III-A–III-D). The α-iduronidase sequence from *Mus musculus* was utilized as an outgroup. β-xylosidases that were previously characterized biochemically were phylogenetically affiliated with either Class II (*Bacillus halodurans* (BAB04787.1) and *Geobacillus stearothermophilus* (ABI49941.1) in bottom Firmicutes wedge, and *Thermoanaerobacterium saccharolyticum* (AAB68820.1) in middle Firmicutes wedge) or Class I (*Caulobacter crescentus* (ACL95907.1), bottom α-Proteobacteria wedge). Bgxg1, from *Orpinomyces* sp. strain C1A, is shown highlighted in yellow.

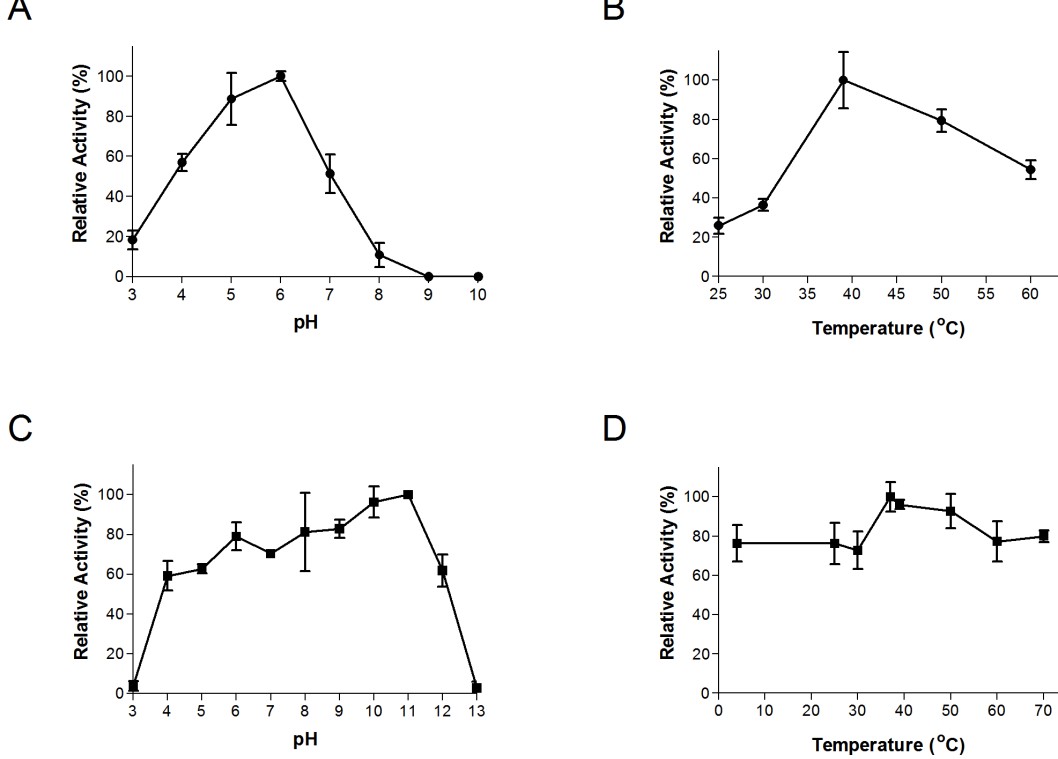

**Figure 2** **Effect of temperature and pH on Bgxg1 activity (A) optimal pH, (B) optimal temperature, (C) pH stability, (D) thermal stability.** All values are presented as relative specific activities, calculated by determining the % activity relative to the highest activity (with the highest activity set at 100%). For (A), (B), (C), and (D), PNPX was used as a substrate. Error bars represent standard deviation of triplicate ($n = 3$) samples.

(Figs. 2A and 2B). The thermal and pH stabilities of Bgxg1 were examined by conducting activity assays post-stress (pH or thermal)-incubations as described above. Bgxg1 retained more than 80% of its specific activity post-application of pH stress ranging between 6 and 11 (Fig. 2C), and 60% of its specific activity post application of pH stress of 4, 5, and 12 (Fig. 2C). Further, Bgxg1 retained ≥70% of its specific activity across the broad range of temperature stressors applied (4–70 °C) (Fig. 2D). In addition, exposure to pH stress from 6–11 and temperature stress from 4–70 °C did not produce results that were significantly different from the optimal conditions ($p$-value > 0.05, 95% confidence interval, Fig. 2).

## Substrate specificities and kinetics

To date, all characterized GH39 enzymes exhibit a narrow substrate range ($\beta$-xylosidase or $\alpha$-L-iduronidase) (Table S1). As predicted by sequence analysis, Bgxg1 exhibited $\beta$-xylosidase activity ($11.5 \pm 1.2$ U/mg, Table 1), strong as compared to previously reported $\beta$-xylosidase activities (Table S1). While having a $\beta$-xylosidase activity greater than the majority of the field in Table S1, Bgxg1 does have lower $\beta$-xylosidase activity than other anaerobic fungi *Neocallimastix frontalis* (16 U/mg, *Hebraud & Fevre, 1990*), *Neocallimastix patriciarum* (30.4 U/mg, *Zhu, Cheng & Forsberg, 1994*), *Piromyces communis* (28 U/mg, *Hebraud & Fevre, 1988*), and *Sphaeomonas communis* (27 U/mg, *Hebraud & Fevre, 1988*)

**PeerJ** ______________________________________________

**Table 1  Substrate specificity and specific activity of Bgxg1.**

| Substrate | Activity tested | Specific activity (U/mg ±SD) |
|---|---|---|
| PNPG | $\beta$-glucosidase | 73.4 ± 7.15 |
| Cellobiose | $\beta$-glucosidase | 55.1 ± 5.36 |
| PNPGal | $\beta$-galactosidase | 54.6 ± 2.26 |
| PNPX | $\beta$-xylosidase | 11.5 ± 1.2 |
| Xylobiose | $\beta$-xylosidase | 10.9 ± 0.96 |
| Beechwood Xylan | Xylanase | 10.8 ± 1.25 |
| Avicel | Exoglucanase | ND |
| CMC | Endoglucanase | ND |
| Locust Bean Gum | Mannanase | ND |
| PNPA | Arabinosidase | ND |
| PNPAc | Acetyl Xylan Esterase | ND |
| PNPC | Cellobiohydrolase | ND |
| PNPM | Mannosidase | ND |
| Aldouronic acid | $\alpha$-glucuronidase | ND |

**Notes.**

Abbreviations: PNPC, *p*-nitrophenyl-$\beta$-D-cellobioside; PNPX, *p*-nitrophenyl-$\beta$-D-xylopyranoside; PNPA, *p*-nitrophenyl-$\beta$-D-arabinofuranoside; PNPM, *p*-nitrophenyl-$\beta$-D-mannoside; PNPG, *p*-nitrophenyl-$\beta$-D-glucopyranoside; PNPGal, *p*-nitrophenyl-$\beta$-D-galactopyranoside; PNPAc, *p*-nitrophenyl-acetate.; ND, Not detected.

(Table 1, Table S1); however, these are not of the same GH-family. Interestingly, in addition to $\beta$-xylosidase activity, Bgxg1 also exhibited strong $\beta$-glucosidase (73.4 ± 7.15 U/mg), $\beta$-galactosidase (54.6 ± 2.26 U/mg), and weak xylanase (10.8 ± 1.25 U/mg) activities (Table 1), as compared to reported activities from previously characterized enzymes (Tables S2–S4). Our extensive literature review identified 63 enzymes that have been biochemically-characterized to have $\beta$-glucosidase activity and, of these, only seven have a reported specific activity higher than that of Bgxg1 (Table S2), including *Candida peltata* (108 U/mg, *Saha & Bothast, 1996*), *Thermoascus aurantiacus* (232 U/mg, *Tong, Cole & Shepherd, 1980*), and *Trichoderma reesei* (768 U/mg, *Takashima et al., 1998*). Similarly, we only identified three $\beta$-galactosidase with a reported higher activity than Bgxg1 (Table S3), including *Alicyclobacillus acidocaldarius* (229 U/mg, *Yuan et al., 2008*), *Bifidobacterium adolescentis* (526 U/mg, *Hinz et al., 2004*), and *Thermus aquaticus* (1,750 U/mg, *Ulrich, Temple & Mcfeters, 1972*). On the other hand, the xylanase activity of Bgxg1 is relatively weak, with many previously reported xylanases exhibiting a much higher specific activity (Table S4), such as *Aspergillus niger* (19 U/mg, 25 U/mg, 35 U/mg, and 48 U/mg, *John, Schmidt & Schmidt, 1979*) and *Trichoderma reesei* (46 U/mg, *Takashima et al., 1998*). Bgxg1 exhibited no detectable exoglucanase, endoglucanase, mannanase, arabinosidase, acetyl xylan esterase, cellobiohydrolase, mannosidase, or $\alpha$-glucuronidase activities.

In addition to its high $\beta$-xylosidase, $\beta$-glucosidase, and $\beta$-galactosidase specific activities, Bgxg1 exhibited remarkably high affinities towards all examined substrates, with $K_m$ values (calculated via extrapolation through Lineweaver–Burke plot) in the low nM range for PNPG and PNPGal, the low μM range for PNPX (Table 2, Tables S1–S4).
**Table 2** **Enzyme kinetics for Bgxg1.** $K_m$ and $V_{max}$ values were calculated by extrapolation from Lineweaver–Burke plots.

| Substrate | Activity tested | $K_m$[a] | $V_{max}$ (U/mg) |
|---|---|---|---|
| PNPG | $\beta$-glucosidase | BDL[b] | 769 ± 18 |
| PNPGal | $\beta$-galactosidase | BDL[c] | 769 ± 13 |
| PNPX | $\beta$-xylosidase | 0.00485 mM ± 0.00062 | 127 ± 8 |
| Beechwood Xylan | Xylanase | 0.038 mg/mL ± 0.0039 | 25.6 ± 10 |

**Notes.**

Abbreviations: PNPG, $p$-nitrophenyl-$\beta$-D-glucopyranoside; PNPGal, $p$-nitrophenyl-$\beta$-D-galactopyranoside; PNPX, $p$-nitrophenyl-$\beta$-D-xylopyranoside.

[a]$K_m$ values are expressed in either mM or mg/mL, depending on the substrate tested. Values are shown ± standard deviation of triplicate samples ($n = 3$).

[b]BDL: below detection limit (500 nM). Extrapolated $K_m$ value obtained using Lineweaver–Burke plot was 0.0000125 mM ± 0.0000096.

[c]BDL: below detection limit (500 nM). Extrapolated $K_m$ value obtained using Lineweaver–Burke plot was 0.000214 mM ± 0.000016.

**Table 3** **Substrate competition experiments.** "Activity tested" column refers to the colorimetric substrate tested (PNPX for $\beta$-xylosidase, PNPG for $\beta$-glucosidase) in the presence of the active site inhibitor (cellobiose or xylobiose, at listed Inhibitor concentrations). Specific activity, $K_m$, and $V_{max}$ refer to the values calculated for the colorimetric substrate in each experiment.

| Activity tested | Active site inhibitor | Inhibitor (mM) | Relative specific activity (%) | $K_m$ (mM) | $V_{max}$ (U/mg) |
|---|---|---|---|---|---|
| $\beta$-xylosidase | Cellobiose | 0 | 100 | 0.00485 | 127 |
| | | 10 | 78.9 | 1.438 | 118 |
| | | 20 | 52.1 | 3.51 | 129 |
| $\beta$-glucosidase | Xylobiose | 0 | 100 | 0.0000125 | 769 |
| | | 10 | 75.8 | 0.000235 | 763 |
| | | 20 | 57.2 | 0.00349 | 752 |

## Substrate competition studies

Substrate competition studies were conducted using a variable concentration of an unlabeled substrate (acting as an inhibitor) and a fixed concentration of a chromophore (PNP-based) substrate (Table 3). The results strongly suggest the occurrence of cross-substrate competitive inhibition between xylobiose and cellobiose (Table 3), since the presence of increasing concentrations of a single substrate lowers the specific activity and increases the $K_m$ of the enzyme towards the other substrate, whilst not affecting its $V_{max}$ ($K_m$ and $V_{max}$ calculated via extrapolation through Lineweaver–Burke plot). This pattern strongly indicates that a single active site is responsible for the observed activities (Table 3), a conclusion that is in agreement with the lack of identifiable additional domains other than pfam01229 in Bgxg1, as well as with the structural modeling data described below.

In single substrate assays, Bgxg1 was capable of converting cellobiose to glucose and xylobiose to xylose at a very fast rate (Figs. 3A and 3B). This reaction occurs more quickly for xylobiose, as a stable maximal xylose concentration is reached after only 1 min of incubation (Fig. 3B), compared to 15 min for glucose release from cellobiose (Fig. 3A). However, the extent of sugar release at the conclusion of the experiment was higher in

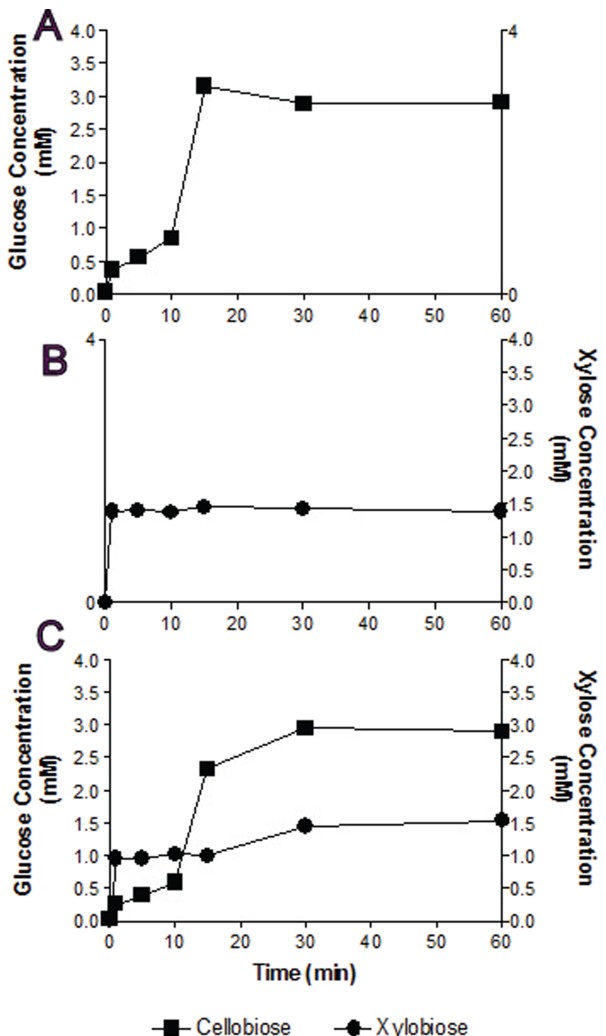

**Figure 3** **Substrate competition and Bgxg1 preference.** Monosaccharides (glucose (■) or xylose (●)) release was assayed when Bgxg1 was challenged with 10 mM cellobiose (A), 10 mM xylobiose (B), or an equimolar mixture of both substrates (C). In (A), the effect of xylobiose (as a competitive inhibitor) is measured through conducting a $\beta$-glucosidase activity assay. In (B), the effect of cellobiose (as a competitive inhibitor) is measured through conducting a $\beta$-xylosidase activity assay. In (C), a competition assay was performed with both cellobiose and xylobiose present, assaying for the presence of glucose or xylose.

cellobiose incubations (Fig. 3A) than xylobiose incubations (Fig. 3B). Competition studies using equimolar concentrations of both substrates revealed the preference of Bgxg1 for xylobiose, since a higher proportion of xylose rather than glucose was detected within the first 15 min of the incubation (Fig. 3C). Nevertheless, the final concentrations of sugars released after 60 min of incubation did not differ when comparing single substrate versus competition experiments (Figs. 3A–3C). Similar to the patterns observed in single substrate assays, Bgxg1 reduced a larger amount of cellobiose to glucose than xylobiose to xylose in competition experiments (Fig. 3C), which is consistent with the higher affinity (lower $K_m$ value) of Bgxg1 for PNPG (12.5 nM) over PNPX (4.85 µM) (Table 2).

## Structure activity predictions

The Bgxg1 protein sequence was submitted to I-TASSER for structural prediction by Iterative Threading ASSEmbly Refinement (*Roy, Kucukural & Zhang, 2010*; *Yang et al., 2015*; *Zhang, 2008*) utilizing the $\beta$-xylosidase originating from *Thermoanaerobacterium saccharolyticum* (PBD entry 1UHV) as a template for model creation (*Roy, Kucukural & Zhang, 2010*; *Yang et al., 2004*; *Yang et al., 2015*; *Zhang, 2008*). Bgxg1 is predicted to have three distinct domains: a catalytic $(\alpha/\beta)_8$ barrel fold domain (position 26–307), a small $\alpha$-helical domain (position 1–25), and a $\beta$ sandwich domain (position 308–344) (Fig. S4). Overall, the structure is predicted to contain 11 $\beta$-sheets (8 in $(\alpha/\beta)_8$-barrel, 3 in $\beta$-sandwich), and 10 $\alpha$-helices (8 in $(\alpha/\beta)_8$-barrel, 2 in $\alpha$-domain). The catalytic $(\alpha/\beta)_8$-barrel fold domain is predicted to consist of eight parallel $\beta$-sheets ($\beta$1–$\beta$8), and eight parallel $\alpha$-helices ($\alpha$1–$\alpha$8). Consistent with $\beta$-xylosidases of *Thermoanaerobacterium saccharolyticum* (1UHV) and *Geobacillus stearothermophilus* (1PX8), the active site pocket of Bgxg1 is predicted to be located on the upper side of the $(\alpha/\beta)_8$-barrel (*Czjzek et al., 2005*; *Yang et al., 2004*) (Fig. S4). Alignment and structural predictions identified the conservation of the general acid–base active site residue Glu127 in the C-terminal of $\beta$3, as part of the GH39-conserved Asn126-Glu127-Pro128 motif as well as the nucleophilic residue Glu225 in $\beta$6 (Fig. 4) (*Czjzek et al., 2005*; *Yang et al., 2004*).

Using the predicted model we sought to infer structural differences potentially responsible for the observed relaxed substrate specificities in Bgxg1 by investigating the amino acid conservation patterns between Bgxg1 and all structurally and/or biochemically-characterized $\beta$-xylosidases. These enzymes are: *Thermoanaerobacterium saccharolyticum* $\beta$-xylosidase (*Yang et al., 2004*), *Geobacillus stearothermophilus* $\beta$-xylosidase (*Bhalla, Bischoff & Sani, 2014*; *Czjzek et al., 2005*), and *Bacillus halodurans* C-125 protein BH1068 (*Wagschal et al., 2008*), all of which belong to Class II (Fig. 1), as well as *Caulobacter crescentus* CcXynB2 (*Correa et al., 2012*), which belongs to Class I (Fig. 1). All of these enzymes have previously been reported to possess $\beta$-xylosidase activity (*Bhalla, Bischoff & Sani, 2014*; *Correa et al., 2012*; *Czjzek et al., 2005*; *Wagschal et al., 2008*; *Yang et al., 2004*). We focused on 25 amino acids in two groups: (i) those previously shown to be important for $\beta$-xylosidase activity (*Czjzek et al., 2005*; *Yang et al., 2004*) (this group includes (in addition to the conserved general acid–base and nucleophilic active sites described above) amino acids providing the tight hydrogen bonding necessary to stabilize the xylosyl-enzyme intermediate formed during the reaction, such as Arg52, His54, Asn159, His228, Tyr230, Glu278, Trp315, Glu322, and Glu323 (locations refer to position in 1UHV)), as well as (ii) those physically interacting with the active site as deduced by the predicted Bgxg1 model (Fig. S3A) (this group includes Val46, Val81, Ile124, Trp125, Gly130, Thr131, Trp132, Phe139, Pro162, Cys163, Tyr164, Ser165, Lys171, His192, Asn242, and Lys247 (locations refer to position in Bgxg1)). Of these 25 amino acids, 15 differed between Bgxg1 and the four other proteins. Five of these 15 amino acids were not conserved amongst any of the five sequences studied and so were not further investigated (Fig. 4). Therefore, 10 distinct differences (eight substitutions and two deletions) between Bgxg1 on one hand and the four biochemically-characterized $\beta$-xylosidases on the other were identified (Table 4).

**Figure 4  Alignment of Bgxg1 and the four biochemically-characterized GH39-family enzymes, highlighting structural predictions and conservation of or around the active site.** Structural predictions for Bgxg1 sequence were obtained using I-TASSER three-dimensional model (Fig. S3) (*Roy, Kucukural & Zhang, 2010*; *Yang et al., 2015*; *Zhang, 2008*). Bgxg1 sequence is compared to those from *Caulobacter crescentus*, *Thermoanaerobacterium saccharolyticum*, *Geobacillus stearothermophilus*, and *Bacillus halodurans*. α-helices in blue are those within the small α-helical domain, α-helices and β-sheets in green are those within the (α/β)₈ barrel, and β-sheets in red are those within the β-sandwich. Red stars (*) represent catalytic residues within the active site. Black stars (*) represent those residues close to the active site, as determined within the Bgxg1 model. Blue stars (*) represent residues noted in the literature to be important for β-xylosidase function (*Czjzek et al., 2005*; *Yang et al., 2004*).

These differences that are predicted to exist in or around the active site of Bgxg1 would putatively impact the size, charge, and/or polarity within the active site (Table 4, Fig. S4).

The expanded substrate specificity observed in this study could be a unique trait in Bgxg1, or it could be specific to all GH39 CAZymes of anaerobic fungi (e.g., Class III-C), or to the entire Class III $\beta$-xylosidases. Based on the above speculations about the amino acids potentially responsible for Bgxg1 relaxed specificity, we further investigated the conservation of these 10 amino acid changes (Table 4) within class III of GH39 proteins. Bgxg1 (as well as other GH39 proteins encoded in C1A genome), all three GH39 proteins from the *Piromyces* genome (accession numbers shown in Fig. 1), and all additional sequences from Class III-C belonging to the genera *Clostridium* and *Teredinibacter* were found to encode 9 of the 10 observed amino acid substitutions (Table 4). However, within the broader Class III, little similarity in key amino acids was observed between Bgxg1 sequences and $\beta$-xylosidases belonging to Class III-A, III-B, or III-D (Table 4). Collectively, these results putatively suggest that the observed relaxed specificity in Bgxg1 could be exclusive to Class III-C $\beta$-xylosidases.

## DISCUSSION

In this study, we used a transcriptomics-guided approach to identify, clone, express, and characterize a GH39 protein (Bgxg1) from the anaerobic gut fungus *Orpinomyces* sp. strain C1A. Our results demonstrate that the expressed protein is multifunctional, possessing strong $\beta$-xylosidase (11.5 U/mg), $\beta$-glucosidase (73.4 U/mg), and $\beta$-galactosidase (54.6 U/mg) activities, as well as a weak xylanase activity (10.8 U/mg) (Tables 1 and 2), as compared to previously characterized enzymes (Tables S1–S4). This novel multi-functionality has not been previously reported in GH39 enzymes (*Bhalla, Bischoff & Sani, 2014*), and therefore this work expands on the known activities of GH39 CAZyme family. Further, Bgxg1 retains high levels of activity over a wide range of temperatures (>80% of activity retained between 4–70 °C) (Fig. 2D) and pH values (>80% of activity retained between pH 6–11) (Fig. 2C). Though the composition of commercial enzymes cocktails are largely proprietary, the presence of 80–200 different components within a mixture has been previously reported (*Banerjee, Scott-Craig & Walton, 2010*; *Van Dyk & Pletschke, 2012*). It is intuitive to think that the inclusion of such a large number of enzymes represents a large contribution to the cost of production. It is here that Bgxg1 would be beneficial, as the inclusion of a single enzyme, possessing multiple strong activities, would lower the cost of production in biorefineries and therefore would be beneficial to the bottom line.

In addition to its relaxed substrate specificity, the enzyme displays strong kinetic properties (high specific activity and affinity) towards its multiple substrates ($K_m$ and $V_{max}$ values calculated via extrapolation through Lineweaver–Burke plot). As a $\beta$-xylosidase, Bgxg1 has one of the highest $\beta$-xylosidase specific activity among all reported ambient (<50 °C) $\beta$-xylosidases, but lower than other anaerobic fungi *Neocallimastix frontalis* (16 U/mg, *Hebraud & Fevre, 1990*), *Neocallimastix patriciarum* (30.4 U/mg, *Zhu, Cheng & Forsberg, 1994*), *Piromyces communis* (28 U/mg, *Hebraud & Fevre, 1988*), and *Sphaeomonas communis* (27 U/mg, *Hebraud & Fevre, 1988*) (Table 1, Table S1). Bgxg1 also one of the

Morrison et al. (2016), *PeerJ*, DOI 10.7717/peerj.2289

**Table 4** Comparison of key amino acids between Bgxg1 and all four biochemically characterized (BC) β-xylosidases from *Thermoanaerobacterium saccharolyticum, Bacillus halodurans, Geobacillus stearothermophilus, Caulobacter crescentus*, as well as in Classes III-A, III-B, and III-D.[a]

| Pos.[b] | AA in Bgxg1 | AA in 4 BC[d] | Significance of change | Importance of residue | Class III-A | Class III-B | Class III-C | Class III-D |
|---|---|---|---|---|---|---|---|---|
| 46 | Val | Tyr | Small, nonpolar (Val) vs. Large, polar (Tyr) | Near active site | NC | Ile | Val | NC |
| 129 | Asp | Asn | Negative charge (Asp) vs. Neutral charge (Asn) | H-bonding | Lys | Asp | Asp | Asp |
| 131[c] | Thr/NC | Phe | Small, polar (Thr) vs. Large, nonpolar (Phe) | Near active site | NC | NC | NC | NC |
| 139 | Phe | Tyr | Large, nonpolar (Phe) vs. Large, polar (Tyr) | Near active site | Tyr | Tyr | Phe | Tyr |
| 163 | Cys | Ala | Polar, thiol (Cys) vs. Nonpolar (Ala) | Near active site | Tyr | Ala | Cys | Tyr |
| 171 | Lys | Trp | Positive charge (Lys) vs. Nonpolar (Trp) | Near active site | Trp | NC | Lys | Lys |
| 194 | Leu | Tyr | Small, nonpolar (Leu) vs. Large, polar (Tyr) | H-bonding | Ser | Ile/Glu | Leu | Tyr |
| 242 | Arg | Ala | Positive charge (Arg) vs. Small, nonpolar (Ala) | Near active site | NC | NC | Arg | NC |
| 322–323 | -gap- | Glu | Gap vs. Negative charge (Glu) | H-bonding | Arg/Thr/Lys | -gap- | -gap- | -gap- |
| 322–323 | -gap- | Glu | Gap vs. Negative charge (Glu) | H-bonding | Gly/-gap- | -gap- | -gap- | -gap- |

Notes.

NC, Not conserved.

[a]No changes were identified in 10 different positions (Arg48, Ile124, Trp125, Asn126, Glu127, Pro128, Trp132, Pro162, His192, Glu225), and 5 positions were variable across all sequences (Val81, Gly130, Tyr164, Ser165, Lys247).

[b]Pos. (Positions) refer to the position of the amino acid in Bgxg1.

[c]Bgxg1 and all proteins in Class III-C β-xylosidases have identical amino acid sequences in all key positions with one exception (Thr131).

[d]Sequences identified using the alignment presented in Fig. 4.

highest specific activities amongst known GH39 $\beta$-xylosidases (Table 1, Table S1), with a lower specific activity than the thermophilic *Thermoanaerobacterium saccharolyticum* (53.8 U/mg, *Shao et al., 2011*) and the thermophilic *Geobacillus stearothermophilus* (133 U/mg, *Bhalla, Bischoff & Sani, 2014*). Compared to other characterized $\beta$-glucosidases, Bgxg1 has the highest specific activity for all ambient temperature $\beta$-glucosidases, and one of the highest reported specific activities among all $\beta$-glucosidase (members of GH1, GH3, GH5, GH9, and GH30 (*Cairns & Esen, 2010*)), regardless of optimal temperature and GH affiliation (Table 1, Table S2). Finally, compared to other characterized $\beta$-galactosidases, Bgxg1 has the highest specific activity for all ambient temperature $\beta$-galactosidases, and one of the highest reported specific activities among all $\beta$-galactosidases (members of GH1, GH2, GH35, and GH42 (*Skalova et al., 2005*)) regardless of optimal temperature and GH affiliation (Tables 1, 2, Table S4).

We reason that the observed kinetics and substrate specificity of Bgxg1 are beneficial for strain C1A and are highly desirable for a saccharolytic enzyme acting within the highly competitive rumen environment, where strain C1A originally existed (*Orpinomyces* sp. strain C1A was isolated from the feces of an angus steer (*Youssef et al., 2013*)). The high specific activity and high substrate affinity may aid in fast and efficient scavenging of sugars from the surrounding environment, where competition for sugars/oligosaccharide produced by saccharolytic enzymes are intense, and where free sugar levels are permanently low (*Garcia-Vallve, Romeu & Palau, 2000*). We hence speculate that the survival in an anaerobic, eutrophic, and highly competitive environment might be responsible for the acquisition, retention and directed evolution of anaerobic fungal $\beta$-xylosidases towards superior kinetics and relaxed specificities.

Sequence analysis and structural predictive modeling (Fig. 4, Fig. S4), and substrate competition experiments (Table 3) predict the presence of a single conserved active site within the $(\alpha/\beta)_8$-barrel fold structure typically observed in GH39-family enzymes (*Czjzek et al., 2005*; *Yang et al., 2004*) (with the conserved catalytic nucleophile (Glu225) and general acid–base residue (Glu127)) and potentially mediating all observed hydrolytic activities). To provide clues regarding the structural basis of the observed multi-functionality, comparison of amino acid conservation patterns putatively affecting the active site topology between Bgxg1 and biochemically characterized GH39 xylosidases, all four of which display no additional activities beyond $\beta$-xylosidase, was undertaken. We identified ten different distinct amino acid changes (8 substitutions and 2 deletions) (Table 4, Figs. S4 and S5) in Bgxg1 that putatively affect the polarity (Tyr vs. Val46, Phe vs. Thr131, Tyr vs. Phe139, Ala vs. Cys163, Trp vs. Lys171, Tyr vs. Leu194, and Ala vs. Arg242), constitute significant size changes (Tyr vs. Val46, Phe vs. Thr131, Tyr vs. Leu194, and Ala vs. Arg242), result in the addition of charged moieties or unique functional groups (Asn vs. Asp129, Ala vs. Cys163, Trp vs. Lys171, and Ala vs. Arg242), or result in the deletion of a negatively charged residue, previously determined to be important (Glu322-323 vs. deletion) to the active site (*Czjzek et al., 2005*). The impact of these speculated changes is unclear, and it remains to be seen if any, all, or a combination of the above differences is responsible for the observed relaxed specificity. However, while all these amino acid changes are speculated to theoretically explain the relaxed specificity of Bgxg1, one such

difference is peculiar and deserves special scrutiny; deletions/gaps in the Bgxg1 sequence as opposed to negatively charged glutamic acids in the other four sequences (Table 4, Fig. S4S). GH39 enzymes belong to the wider family of $\beta$-1,4-retaining hydrolases of clan GH-A e.g., GH1 $\beta$-glucosidase and GH5 cellulases. Differences in structure between $\beta$ 1,4-glucose cleaving enzymes and $\beta$ 1,4-xylose cleaving enzymes within clan GH-A have been extensively investigated (Czjzek et al., 2005; Czjzek et al., 2001; Ducros et al., 1995; Hovel et al., 2003; Verdoucq et al., 2004). Such studies have demonstrated that, within the active site of $\beta$ 1,4-glucose cleaving enzymes, a Gln residue (corresponding to position 39 in the enzyme dhurinase of *Sorghum bicolor* (Czjzek et al., 2005; Ducros et al., 1995; Verdoucq et al., 2004)) interacts with the substrate by forming a hydrogen bond with O3 and O4 of the glucose moiety (Czjzek et al., 2005; Ducros et al., 1995). On the other hand, $\beta$ 1,4-xylosidases acting on C5 sugar dimers contain a Glu residue in lieu of Gln (at position 322–323 in *Thermoanaerobacterium saccharolyticum*, Fig. 4, Fig. S4, Table 4) that binds to O3 and O4 of the xylose moiety (Czjzek et al., 2005). Interestingly, these Glu residues are aligned with a gap in the sequence of the multifunctional Bgxg1 (Fig. 4), with no apparent occurrence of either Glu or Gln amino acids within the vicinity. Structurally predictive modeling suggests that in lieu of these Glu322-323 residues (1UHV numbering) Bgxg1 is predicted to possess Gly-Arg at an approximately sterically-similar location near the active site (Fig. S4R-S), representing a significant change from two negatively-charged residues, to an uncharged and positively-charged pair of residues. Since the Glu residues in biochemically characterized $\beta$-xylosidases are shown to be important for stabilizing intermediates (Czjzek et al., 2005), the predicted absence of these residues in Bgxg1 and their speculated replacement with Gly-Arg suggests that Bgxg1 might employ a different mechanism for stabilizing its intermediates during the catalytic process; however, this speculation will require further investigation.

The ecological relevance, global distribution, and evolutionary patterns of multi-functionality within GH39 $\beta$-xylosidases remain to be conclusively determined. Phylogenetic analysis demonstrated the occurrence of nine out of ten amino acids substitutions/deletions in all sequenced members of Class III-C, residues which we speculate to be of importance to the observed multi-functionality of Bgxg1, but as Bgxg1 is the only biochemically-characterized enzyme within Class III, this analysis is purely speculative (Table 4, alignment in Fig. S5). In addition to anaerobic fungal sequences, Class III-C $\beta$-xylosidases contain sequences from the genera *Clostridium* and *Teredinibacter* (Fig. 1). Since it has been previously demonstrated that the xylanolytic machinery in anaerobic fungi, including $\beta$-xylosidases, has been acquired from bacteria via horizontal gene transfer (Youssef et al., 2013), and speculation that some or all of the amino acids substitutions/deletions in members of class III-C collectively account for the observed multi-functionality (though it is unknown, at this time, whether these GH39 enzymes possess this multi-functionality), we therefore reason that the observed distribution pattern suggests the evolution of relaxed specificity in GH39 $\beta$-xylosidases within the domain Bacteria, prior to the acquisition of GH39 $\beta$-xylosidases by the anaerobic fungi and that the acquired capability is speculated to be retained in all anaerobic fungal GH39 $\beta$-xylosidases.

## CONCLUSIONS

In conclusion, we have characterized a novel $\beta$-xylosidase that represents the first GH39-family enzyme cloned and expressed from anaerobic fungi. The enzyme is multi-functional, capable of hydrolyzing cellobiose, xylobiose, as well as several PNP-glycosides. It also displays high affinity towards various substrates, retains activity over a wide range of temperatures and pHs, and possesses excellent temperature and thermal stability. Structurally predictive modeling identified putative differences which potentially could account for the observed relaxed specificity. Collectively, these capabilities render Bgxg1 an excellent candidate for inclusion in enzyme cocktails mediating cellulose and hemicellulose saccharification from lignocellulosic biomass (*Morrison, Elshahed & Youssef, in press*).

## ACKNOWLEDGEMENTS

We thank Dr Gilbert John (Oklahoma State University) for supplying the *E. coli* BL21(DE3)pLysS cells used in this study. We also thank Dr Robert Gruninger for helpful discussions.

### Funding

This work was supported by the Department of Transportation Sun Grant Initiative award number DTOS59-07-G-00053. The funders had no role in study design, data collection and analysis, decision to publish, or preparation of the manuscript.

### Grant Disclosures

The following grant information was disclosed by the authors:
The Department of Transportation Sun Grant Initiative: DTOS59-07-G-00053.

### Competing Interests

The authors declare there are no competing interests.

### Author Contributions

- Jessica M. Morrison conceived and designed the experiments, performed the experiments, analyzed the data, wrote the paper, prepared figures and/or tables, reviewed drafts of the paper.
- Mostafa S. Elshahed conceived and designed the experiments, contributed reagents/materials/analysis tools, wrote the paper, reviewed drafts of the paper.
- Noha Youssef conceived and designed the experiments, contributed reagents/materials/analysis tools, reviewed drafts of the paper.

### DNA Deposition

The following information was supplied regarding the deposition of DNA sequences:
GenBank accession number KT997999.

## Data Availability

The raw data has been supplied as Data S1.

## Supplemental Information

Supplemental information for this article can be found online at http://dx.doi.org/10.7717/peerj.2289#supplemental-information.

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
