# Peer review of "A multifunctional GH39 glycoside hydrolase from the anaerobic gut fungus Orpinomyces sp. strain C1A"

_PeerJ, doi:10.7717/peerj.2289_

## Round 0.1 · original submission · Minor Revisions

While all reviewers agreed that the paper was well-written with important conclusions, they also agreed that minor revisions are necessary prior to consideration for publication, and they have extensive comments and suggestions to improve the manuscript.

·

Basic reporting

No Comments

Experimental design

No Comments

Validity of the findings

The pH stability data is of interest considering that enzymes with a wide pH stability range are currently needed for potential applications with alkaline ionic liquids for combined pretreatment-saccharification. However, data of activity assays in diverse buffers would be more relevant to determine stability than the pretreatment assays shown. It is possible that the pretreatment assays shown may not accurately reflect stability of the enzyme to function in diverse pH conditions.
In my opinion some of the claims on the properties of the enzyme and its potential use in biorefineries need to be toned down. The new enzyme has some interesting and unique characteristics, but it displays similar pH and temperature stability and has lower activity than other previously reported enzymes (as shown in the supplementary material). Consequently, some of the language used seems to "oversell" the qualities of the enzyme. For instance, in line 407: "enzyme displays superior kinetic..", what is Bgxg1 superior to? in lines 409-411, "one of the highest" is used, yet this does not specify how much lower activity than other enzymes Bgxg1 displays. In fact, it is difficult to asses if the comparisons in the supplementary tables are accurate: are all these enzymes assayed using the same type of substrate/technique/experimental conditions? If so, there are enzymes that are substantially more active than Bgxg1 for each substrate and they would probably be preferably used in biorefineries over Bgxg1 (in disagreement with the statement in lines 492-494). It seems the multiple activities present in Bgxg1 may be the most unique feature of this enzyme, but it is unclear if any of the other previously described enzymes display similar wide range substrate specificity.

Additional comments

Very well written manuscript, easy to follow rationale for the different experiments.

Reviewer 2 ·

Basic reporting

This manuscript documents the in vitro enzyme activity and substrate specificity of a GH 39 xylanase cloned from the anaerobic gut fungus, Orpinomyces sp. strain C1A. Overall, the authors provide a wealth of background information in the introduction and methods that enables reviewers from a range of disciplines to demonstrate why this work is important. However, several minor details could be added to the introduction and methods for additional clarity:


Intro Line 95: Which herbivorous guts are these fungi commonly found in association with? Also, some background information about the strain that was used for this study would be informative.
Introduction: The authors should more concisely summarize their findings in the introduction or just present the findings in the results and discussion sections.
Methods: Could the authors provide the accession number to the WGS submission for the genome so that others who are interested can locate contig ASRE01002650 and gene ctg7180000059688?
Methods: For the IMG accession, could the authors also provide the url for the IMG entry for this fungal isolate?
Methods: The authors include the codon optimized sequence in a supplemental file. It would be beneficial to include the optimized codon sequence aligned to the actual sequence in a supplemental file, highlighting which bases were changed for expression in E. coli.
Methods: For the DNS assay, what substrate(s) was used for the standard curves?
Methods: How were the PNPs released from cellobiohydrolase, β-xylosidase, arabinosidase, mannosidase, β-glucosidase, β-galactosidase, and acetyl xylan esterase detected?

Experimental design

The objectives and outcomes of the majority of the experiments are very clearly presented and the majority of the experiments presented by the authors support their findings. However, there are several areas that could be improved upon:

Results (Lines 212-213) and Methods (Lines 212-214): A little more explanation is needed for the pH and temperature stress treatments performed to determine enzyme activity after prolonged exposure to various pH and temperature extremes. After the enzymes were held at various temperatures or at various pHs for the one hour time period, what were the reaction conditions used to assay enzyme activity? Were the activities measured at the same pH/temperatures that they were held at or were the assays performed at the pH/temperature optima determined in Figure 3A/B? I also recommend adding a description for how relative activity was calculated, particularly for Figure 3C and D, in the figure legend. The substrate used for these experiments should also be mentioned in the figure legend. Interestingly, it appears as though the enzyme activities are relatively unchanged after exposure to various pHs and temperatures, suggesting the enzyme is stable wide range of temperatures and pHs. Were any statistical analyses performed to identify pH and temperatures where the enzyme activity either deviated or did not deviate significantly from the activity observed under optimal conditions? The outcome of these statistical tests are necessary to show whether or not the activities were significantly altered.

Methods 242-246 and Results (Lines 315-317): Why were Lineweaver-Burke plots used to estimate Vmax and Km in this study? These plots can be error prone and can distort Km and Vmax values. Eadie–Hofstee plots can be a bit more robust. Therefore, the high Km values presented by the authors in the results section should be qualified, especially given that the values for PNPG and PNPGal were extrapolated based on the plots. Additionally, in Table 2, results for Km are presented as mM for PNPG, PNPGal, and PNPX are presented as mM, but beechwood xylan is presented as mg/mL. Is that correct? I also suggest adding some additional details to the legend for Table 2 to indicate briefly how these values were determined.

Validity of the findings

There are several areas in the results and discussion sections where I feel the authors have overspeculated their findings, given what is currently known about G 39 enzymes.

In particular, the use of Lineweaver-Burke plots to calculate Km and Vmax may result in distorted values, particularly for values that were extrapolated (PNPG and PNPGal). Therefore, the high beta-glucosidase and high beta-galactosidase activity levels relative to previously characterized enzymes could be an artifact of this analysis. In addition, these values were extrapolated, which can add further noise to these calculations.

Results Lines 366-367: The authors claim that all of the enzymes presented solely possess beta-xylosidase activity. However, in looking at the data presented in these manuscripts, it does not appear that Bhalla et al, Correa et al, Czjzek et al, or Yang et al tested multiple substrates to determine whether or not these enzymes possessed additional activity beyond beta-xylosidase. Wagschal et al examined the ability to hydrolyze the following substrates: β-d-xylopyranose, α-l-arabinofuranose, and α-l-arabinopyranose and was able to conclude that the GH 39 in this study was able to hydrolyze all of these substrates, but that it was inactive on arabinan, and released only xylose from oat, wheat, rye, beech, and birch arabinoxylan (and thus, it likely could be classified as a xylosidase). Therefore, the authors should state that the ability of the beta-xylosidases presented in these other studies to hydrolyze other substrates is unknown (unless there are additional references that point to their specific activities; if so, please add them). Additionally, the relevance in the residues identified as being linked to the expanded substrate specifities of Bgxg1 should also be qualified by the authors since it seems to be unknown whether any of the beta-xylosidases used in the comparison could also be multifunctional.

Results Lines 392-394: An additional supplemental figure depicting an alignment of Bgxg1 with other Class C GH 39 enzymes would be helpful to identify additional residues shared between these enzymes beyond the 10 depicted in Table 4. This would be helpful given that the evidence presented by the authors to demonstrate that Bgxg1 is the only enzyme in the comparative analysis capable of hydrolyzing multiple substrates is not sufficient. This could led to the identification of other shared residues in this class that could be candidates for any unique activity associated with this class of enzymes.

Discussion (Lines 402-403): change ‘not previously encountered’ to ‘not previously reported’ It is likely that previously encountered GH 39 enzymes are also multifunctional, but their activity across multiple substrates has not been reported.

Discussion (Line 436): Has the unifunctionality of these beta-xylosidases been verified? See my comment for results lines 366-367.

Discussion (Lines 479-484): The speculations presented in these passages need to be modified or qualified in some manner. First, it is unknown whether or not the GH 39 enzymes in this clade have the observed multifunctionality observed in Bgxg1. These enzymes could have undergone significant functional divergence since their appearance in the genome of the LCA of the anaerobic fungi. Second, the residues identified by the authors have not been conclusively linked to the multifunctionality observed in Bgxg1. The evidence presented in the manuscript suggests that these residues are unique to a certain clade of anaerobic fungal GH 39 enzymes, but only one has been shown to have multiple functions and the authors have not presented sufficient evidence in the manuscript to demonstrate that the GH 39 enzymes presented in other studies exclusively act as beta-xylosidases and could not catalyze the same types of reactions as Bgxg1.

Additional comments

Some minor editorial comments for your consideration:

Abstract Line 53: remove “strength determined”
Methods (Line 243-246): I think there is a word missing from this sentence. “using double-reciprocal Lineweaver-Burke plots, which were used…”
Results (Lines 301-314): It would be helpful for the authors to point out a couple of examples from Tables S2-S4 in the text to support their claims that Bgxg1 exhibits strong beta-glucosidase and beta-xylosidase activities, but weak xylanase activities compared to previously characterized enzymes.
Figure 4: More information is needed for this figure legend to determine what data is being presented in panels A, B, and C
Discussion (Line 423): may aid in fast and efficient scavenging
Discussion (Line 426): anaerobic, eutrophic, and highly competitive environment
Conclusion Line 490: Structural predictive modeling

Reviewer 3 ·

Basic reporting

Figure 2 is unnecessary.

Experimental design

No comments

Validity of the findings

No comments

Additional comments

General Comments: The manuscript by Morrison and colleagues describes the preliminary characterization of a multifunctional glycoside hydrolase from CAZy family GH39. The novelty of this study is related to the characterization of a GH39 family enzyme from an anaerobic fungus and the broad substrate range exhibited by this enzyme. The paper is well written. The study objectives are clearly defined and the work is well justified although perhaps a bit overstated regarding the connection to biofuel production. The methods are appropriate and most conclusions well supported although there are a few concerns that need to be addressed in the specific comments below. My major concern about the work is that predictions were made regarding active site and residues potentially contributing to the relaxed specificity of Bgxg1 and yet these predictions were not tested in the laboratory. I feel the lack of these experiments greatly weakens the significance of this work.

Specific comments

Line 53. “2.26 U/mg, respectively) and a weak xylanase activity (10.8 ± 1.25 U/mg), strength determined as compared to previously characterized enzymes.” – I don’t believe “strength determined” is needed in this sentence.

Line 96. “Neocallimastigomycota are restricted to the herbivorous gut, where they are responsible for the…” Is this group really only restricted to the “herbivorous gut”? I also question the references cited in support of this claim. These references largely focus on fiber degrading enzymes from anaerobic fungi not the biology of these microbes.

Line 107 + 108. I am not sure that I fully agree with the authors’ assessment of the paucity of information on anaerobic fungal enzymes. I think they need to do a more thorough review on the subject.

Lines 152 - 155. Was the “entire” expression construct synthesized or rather was the bgxg1 insert synthesized and cloned into pET28a?

Lines 243 - 246. Something is missing from sentence. It looks like it should be split into two sentences.

Line 266. “were” should be “where”

Figure 2. Is this figure necessary?

Figures 3A and B. I would have preferred to see the effect of pH and temperature assessed every 0.5 pH units and 5 degrees C, respectively.

Line 298. According to Figure 3D, on average Bgxg1 retains > 70% of its specific activity rather than >80% of its specific activity.

Line 302. Is “Predictably” appropriate? How was this predicted?

Lines 333-336. According to Fig 4C, it appears that more glucose is released in the first 15 minutes but within the first minute, approximately 4 times the amount of xylose is released. I also think the legend at the bottom of the page of Fig 4A-C should be glucose and xylose rather than cellobiose and xylobiose.

Lines 355 – 358. Why not test the active site predictions by mutating Glu127 and Glu225?

Lines 384 - 395. Why not test the predictions regarding the relaxed specificity for Bgxg1?

---

## Round 0.2 · accepted · Accept

I appreciate your efforts to address the lengthy comments from the three reviewers, and hope that you agree that the suggestions have greatly improved the manuscript